# Task Confusion and Catastrophic Forgetting in Class-Incremental Learning: A Mathematical Framework for Discriminative and Generative Modelings

**Milad Khademi Nori**[*]
Electrical and Computer Engineering
Queen's University
Kingston, Ontario, Canada
19mkn1@queensu.ca

**Il-Min Kim**
Electrical and Computer Engineering
Queen's University
Kingston, Ontario, Canada
ilmin.kim@queensu.ca

## Abstract

In class-incremental learning (class-IL), models must classify all previously seen classes at test time without task-IDs, leading to task confusion. Despite being a key challenge, task confusion lacks a theoretical understanding. We present a novel mathematical framework for class-IL and prove the Infeasibility Theorem, showing optimal class-IL is impossible with discriminative modeling due to task confusion. However, we establish the Feasibility Theorem, demonstrating that generative modeling can achieve optimal class-IL by overcoming task confusion. We then assess popular class-IL strategies, including regularization, bias-correction, replay, and generative classifier, using our framework. Our analysis suggests that adopting generative modeling, either for generative replay or direct classification (generative classifier), is essential for optimal class-IL.

## 1 Introduction

Incremental learning (IL) has garnered significant interest in academia and industry [1, 2] due to its ability to (i) achieve more resource-efficient learning by avoiding retraining models from scratch with new data, (ii) reduce memory usage by eliminating the need to store raw data, which is vital for complying with privacy regulations, and (iii) develop a learning system that mirrors human learning [3]. There are two categories of incremental learning settings in the literature [4]: (i) task-based [5, 6, 7] and (ii) task-free [8, 9, 10, 11].

Task-based category itself consists of three scenarios [5]: (a) task-incremental learning (task-IL), (b) domain-incremental learning (domain-IL), and (c) class-incremental learning (class-IL). The three scenarios differ at the test time, where the task-IL scenario is given with the task-ID, the domain-IL does not need task-ID to begin with, whereas the class-IL must infer task-ID. The second category, which is task-free, aims to banish the notion of task (boundary) at all, both at the training and test time.

This paper focuses on task-based class-IL, currently the most popular regime [12, 13, 14, 15]. However, our theoretical results and proposed scheme are applicable to task-free settings as well [8, 9]. Progressing towards task-free learning is important because IL, like the brain, should aim to be less reliant on supervision, eliminating the need for task-IDs.

---

[*]Milad's current affiliation is Toronto Metropolitan University, and his current email is mkn@torontomu.ca. His personal email is miladkhademinori@gmail.com.

For incremental learning including class-IL, the main challenge had been thought to be catastrophic forgetting (CF), which *broadly* refers to *any* performance drop on tasks that are previously learned after learning a new one [1]. For class-IL, however, we discourage the usage of this language because it has recently turned out that, in class-IL, not all the performance drop is caused by "forgetting" and indeed most of the performance drop is inflicted by task confusion (TC) [16]. TC originates from the fact that in class-IL, the classes residing in distinct tasks are never seen together; nonetheless must be discriminated among each other at the test time *absent* task-ID (meaning that task-ID must be inferred).

Attributing all performance drops to CF in class-IL is misleading. Using the term "forget" implies that something previously learned is forgotten. However, in class-IL, the model has never learned how to distinguish among tasks because it has never seen those tasks together. At test time, it must infer the task-ID to make these distinctions. Thus, it does not make sense to say that "it forgot how to distinguish among those tasks." In class-IL, we differentiate between performance drops caused by TC and CF.

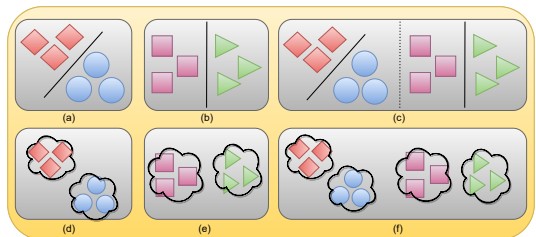

Figure 1: Task Confusion in Discriminative and Generative Modeling.

Despite TC being recently discovered as the main obstacle in class-IL [16, 3], it has not yet been well-understood mathematically/theoretically, and almost all the class-IL works attribute CF, which used to be the only problem in the task-IL scenario, to the performance drop observed in class-IL scenario. And, this is the problem. For that, in Fig. 1 we visualize how TC emerges: when a class-IL model learns tasks 1 and 2 in Figs. 1(a) and 1(b), respectively, it can perform intra-task discrimination in Fig. 1(c); however, it is still incapable of performing inter-task discrimination, whose absence is shown in Fig. 1(c) with a dotted line. This incapability is not because the class-IL model has forgotten the knowledge of task 1 after learning task 2; it is indeed because it has not learned to make inter-task distinction in the first place. In other words, failure in inter-task discrimination has nothing to do with CF; it is about TC which stems from the fact that the class-IL model has never seen classes of different tasks together to be able to make the distinction.

There are two adjacent studies to our work: (i) Kim's [17] and (ii) Soutif-Cormerais' [16]. Although the scope of Kim's work and ours are similar, there are key differences:

- Kim's paper does neither define nor discuss TC. There is not any mention of the term TC throughout the paper, let alone proving its occurrence.
- Kim's work does not prove that in generative modeling TC does not occur while in discriminative modeling it does. There is no mention of discriminative modeling or generative modeling throughout Kim's work.
- Kim's mathematical framework and ours are very different in the sense that their mathematical framework does not appreciate the relationship between TC and discriminative/generative modeling (Comprehensive related work is presented in Appendix).

Also, there are significant contrasts between Soutif-Cormerais' work and ours:

- Soutif-Cormerais' work presents empirical results based on black-box experiments in favor of the importance of cross-task features to mitigate TC. The provided evidence clearly does not prove any statement; rather, it is suggestive. Whereas, our work adopts a white-box approach by breaking down the loss of an N-way discriminative classifier into $\binom{N}{2}$ in Lemma 1 and proving that indeed the optimal performance shall not be obtained unless off-diagonal losses are taken into account. Our approach for studying the role of TC is based on rigorous mathematical analysis.
- Soutif-Cormerais' work is silent on the whereabouts of TC in discriminative/generative modeling. Whereas, our work, by means of rigorous mathematical analysis, proves that while TC does happen in discriminative modeling, it does not in generative modeling [16].

We believe that the essence of TC and CF as well as their distinction have not yet been well-studied from a theoretical perspective. This motivated the current study. In this paper, we present three contributions:

- We introduce a novel mathematical framework that distinguishes between TC and CF in class-IL and task-free settings, clarifying their distinct roles. Unlike existing definitions that often conflate TC and CF based on overall performance, our framework provides a clear distinction.

- Utilizing this framework, we present the Infeasibility Theorem, demonstrating that achieving optimal class-IL in discriminative modeling is impossible even with CF prevention, due to TC. Conversely, we propose the Feasibility Theorem, showing that optimal class-IL is achievable in generative modeling if CF is prevented.

- We further offer corollaries for various class-IL strategies, such as regularization, bias-correction, replay-based methods, and generative classifier schemes, allowing us to discuss their optimality.

In the next section, for discriminative modeling, Lemmas 1 and 2 form the groundwork that leads to Theorem 1 and its subsequent Corollaries 1 through 5. For generative modeling, Lemma 3 underpins Theorem 2 and Corollary 6. Furthermore, Hypothesis 1 is derived from the principles outlined in Lemma 1.

## 2    Mathematical framework for Class-IL

In any incremental learning including class-IL, the goal is to get close to (ideally achieve) the ultimate performance of the non-incremental learning without any forgetting of past tasks and any confusion among tasks, which is obtained when all the data for all tasks are simultaneously available for training. This is a performance upper bound, which will be later denoted as the *joint* scheme in Table 2. To achieve the goal in class-IL, we present our mathematical framework and formulate the training problems to resolve TC and CF. The formulations of TC and CF problems are in themselves meaningful contributions.

In this section, after formulations of TC and CF, we present our first theorem (i.e., the Infeasibility Theorem), where we prove that achieving the optimal class-IL via the implementation of conditional probability $P(Y|X)$ (equivalent to *discriminative* modeling, which is the common practice in the class-IL literature) is essentially infeasible even after preventing CF. Then, in another theorem (i.e., Feasibility Theorem), we prove that achieving the optimal class-IL as an implementation of joint probability $P(X, Y)$ (i.e., *generative* modeling) is feasible.

### 2.1    Discriminative modeling

First, we analyze discriminative modeling: for that, we prove a lemma stating that an implementation of conditional probability (via discriminative modeling) as an $N$-way classifier (with $N$ classes) is equivalent to the implementation of $\binom{N}{2}$ binary classifiers. This lemma will be essential for formulating and understanding the problems of TC and CF.

To that end, we start by defining the classifier's loss function: let $I_{\boldsymbol{\theta}}$ denote the classification error as follows:

$$I_{\boldsymbol{\theta}} = \int_{\mathcal{X} \times \mathcal{Y}} v(f_{\boldsymbol{\theta}}(x), y)p(x, y) \, dxdy \tag{1}$$

where $v(f_{\boldsymbol{\theta}}(x), y)$ is a given loss function, $x, y \in \mathcal{X}, \mathcal{Y}$ are the input data and the label, $f_{\boldsymbol{\theta}}(\cdot)$ denotes the model parameterized by $\boldsymbol{\theta}$, and $p(x, y)$ is the joint probability density function of $(x, y)$. Now, we present our lemma in the following.

**Lemma 1** (Conditional Probability Equivalence Lemma)**:** *An $N$-way discriminative classifier parameterized by $\boldsymbol{\theta}$ implementing conditional probability $P(Y|X)$ with the loss function defined in Eq. 1 is equivalent to the implementation of $\binom{N}{2}$ virtual binary classifiers as follows:*

$$I_{\boldsymbol{\theta}} = \frac{1}{N-1} \sum_{k=1}^{N} \sum_{l=1, l \neq k}^{N} \int_{\mathcal{X}_{kl} \times \mathcal{Y}_{kl}} v(f_{\boldsymbol{\theta}}(x), y)p(x, y) \, dxdy \tag{2}$$

*where $\mathcal{X}_{kl} = \mathcal{X}_k \cup \mathcal{X}_l$, $\mathcal{Y}_{kl} = \mathcal{Y}_k \cup \mathcal{Y}_l$, $\mathcal{X}_k, \mathcal{X}_l \subset \mathcal{X}$, $\mathcal{Y}_k, \mathcal{Y}_l \subset \mathcal{Y}$, $\mathcal{X}_k \cap \mathcal{X}_l = \emptyset$, $\mathcal{Y}_k \cap \mathcal{Y}_l = \emptyset$, for $k \neq l$.*

*Proof: See Appendix A.* □

Simply put, this lemma says that, for example, a 3-way classifier ($N = 3$) can be seen as being made up of 3 underlying binary classifiers ($\binom{N}{2} = 3$). Imagine a 3-way classifier that is supposed to discriminate between cat, dog, and rabbit; this classifier, according to our lemma, can be deemed as having 3 underlying binary classifiers that classify between cat-dog, cat-rabbit, and dog-rabbit.

To understand the implications of Lemma 1, we simplify the notation: we define the loss term of each individual binary classifier of Eq. 2 as follows:

$$\rho_{kl}(\boldsymbol{\theta}) = \frac{1}{N-1} \int_{\mathcal{X}_{kl} \times \mathcal{Y}_{kl}} v(f_{\boldsymbol{\theta}}(x), y) p(x, y) \, dx dy \tag{3}$$

where $\rho_{kl}(\boldsymbol{\theta})$ is the loss corresponding to the virtual binary classifier *discriminating* between classes $k$ and $l$. This enables us to have the overall loss term as simple as follows: $I_{\boldsymbol{\theta}} = \sum_{k=1}^{N} \sum_{l=1, k \neq l}^{N} \rho_{kl}(\boldsymbol{\theta})$. It is worthwhile to view $I_{\boldsymbol{\theta}}$ as a two-dimensional array (matrix) of binary classifiers' loss terms given by

$$\boldsymbol{P}(\boldsymbol{\theta}) = \begin{bmatrix} \oslash & \rho_{12}(\boldsymbol{\theta}) & \dots & \rho_{1N}(\boldsymbol{\theta}) \\ \rho_{21}(\boldsymbol{\theta}) & \oslash & \dots & \rho_{2N}(\boldsymbol{\theta}) \\ \vdots & \vdots & \ddots & \vdots \\ \rho_{N1}(\boldsymbol{\theta}) & \rho_{N2}(\boldsymbol{\theta}) & \dots & \oslash \end{bmatrix} \tag{4}$$

where $\oslash$ denotes 'undefined.' The diagonal losses are undefined because a given class does not have a loss term with itself due to the nature of *discriminative* modeling. In summary, what we have done in Eq. 4 is that instead of saying that we have a single loss function to minimize, for example, for our cat-dog-rabbit classifier, we say that there are three binary classifiers, each with its own loss term; and, then we arranged the losses in the form of a matrix.

Now we consider the task-based (discriminative) class-IL model that sequentially observes $T$ number of tasks, each at a time, with $C$ classes for each task (i.e., $N = T \times C$). The objective is to achieve the performance (as measured by the loss function in Eq. 1) that one would have achieved via having present all $T$ tasks together. We can re-write our loss matrix within *our* system model of class-IL as follows:

$$\boldsymbol{P}(\boldsymbol{\theta}) = \begin{bmatrix} \boldsymbol{P}_{11}(\boldsymbol{\theta}) & \boldsymbol{P}_{12}(\boldsymbol{\theta}) & \dots & \boldsymbol{P}_{1T}(\boldsymbol{\theta}) \\ \boldsymbol{P}_{21}(\boldsymbol{\theta}) & \boldsymbol{P}_{22}(\boldsymbol{\theta}) & \dots & \boldsymbol{P}_{2T}(\boldsymbol{\theta}) \\ \vdots & \vdots & \ddots & \vdots \\ \boldsymbol{P}_{T1}(\boldsymbol{\theta}) & \boldsymbol{P}_{T2}(\boldsymbol{\theta}) & \dots & \boldsymbol{P}_{TT}(\boldsymbol{\theta}) \end{bmatrix}, \boldsymbol{P}_{ij}(\boldsymbol{\theta}) = \begin{bmatrix} \rho_{11}^{ij}(\boldsymbol{\theta}) & \rho_{12}^{ij}(\boldsymbol{\theta}) & \dots & \rho_{1C}^{ij}(\boldsymbol{\theta}) \\ \rho_{21}^{ij}(\boldsymbol{\theta}) & \rho_{22}^{ij}(\boldsymbol{\theta}) & \dots & \rho_{2C}^{ij}(\boldsymbol{\theta}) \\ \vdots & \vdots & \ddots & \vdots \\ \rho_{C1}^{ij}(\boldsymbol{\theta}) & \rho_{C2}^{ij}(\boldsymbol{\theta}) & \dots & \rho_{CC}^{ij}(\boldsymbol{\theta}) \end{bmatrix} \tag{5}$$

in which $\boldsymbol{P}(\boldsymbol{\theta})$ is rewritten as the task-level loss matrix, of which entry is either an intra-task $\boldsymbol{P}_{ii}(\boldsymbol{\theta})$ or inter-task $\boldsymbol{P}_{ij}(\boldsymbol{\theta})$ (for $i \neq j$) loss matrix; and $\rho_{mn}^{ij}(\boldsymbol{\theta})$ is the loss term between the $m$th class of task $i$ and the $n$th class of task $j$. Note that $\rho_{mn}^{ij}(\boldsymbol{\theta}) = \oslash$ for $i = j, m = n$ (See Appendix B for more detail). We will provide a definition, Definition 1, which specifies how such a discriminative class-IL model learns task by task.

Eq. 4 explains that for example if we have four classes, cat, dog, rabbit, and duck, and the first two are in the first task and the second two in the second task, then we have four task-level matrices. The first one, first row and first column, concerns the loss term of the binary classifier discriminating between cat and dog; the second and third loss matrices, the non-diagonal ones, characterize the loss terms of the binary classifiers discriminating between cat-rabbit, cat-duck, dog-rabbit and dog-duck (inter-task binary classifiers); and finally, the last one, second row and column, stands for the loss of the binary classifier discriminating between rabbit and duck.

**Definition 1** (Discriminative Class-IL)**:** *A discriminative class-IL model 'sequentially' trains by minimizing the losses of the diagonal blocks of the loss matrix $\boldsymbol{P}(\boldsymbol{\theta})$ in Eq. 5. That is, a discriminative class-IL model first optimizes $\boldsymbol{\theta}$ by minimizing $|\boldsymbol{P}_{11}(\boldsymbol{\theta})|$, and then re-optimizes $\boldsymbol{\theta}$ by minimizing $|\boldsymbol{P}_{22}(\boldsymbol{\theta})|$, etc, where $|\cdot|$ operator sums up all (defined) components of the given matrix.*

Definition 1 hints at the critical problem; the 'diagonal' is the critical word. The class-IL model only minimizes the diagonal blocks and ignores non-diagonal ones; which is why the TC problem arises: the notorious problem that is misunderstood. This paper clears this misunderstanding.

With Definition 1, now we can also define CF. Properly defining CF is very important since in the literature, CF has been *too broadly* defined based on the overall performance, which causes CF to get conflated with another crucial phenomenon that is TC (will be later defined in Definition 5).

**Definition 2** (Catastrophic Forgetting)**:** *Consider a class-IL model that minimizes the loss $|\boldsymbol{P}_{ii}(\boldsymbol{\theta})|$ of task $i$, achieving the minimal loss $|\boldsymbol{P}_{ii}(\tilde{\boldsymbol{\theta}}_i)|$, where*

$$\tilde{\boldsymbol{\theta}}_i = \operatorname*{argmin}_{\boldsymbol{\theta}} |\boldsymbol{P}_{ii}(\boldsymbol{\theta})|. \tag{6}$$

*Then the model proceeds and minimizes the loss $|\boldsymbol{P}_{(i+1)(i+1)}(\boldsymbol{\theta})|$ of task $(i+1)$, achieving the minimal loss $|\boldsymbol{P}_{(i+1)(i+1)}(\tilde{\boldsymbol{\theta}}_{(i+1)})|$, where $\tilde{\boldsymbol{\theta}}_{(i+1)}$ is given by Eq. 6 with $i$ being replaced by $i+1$. We state that the model committed Catastrophic Forgetting if*

$$|\boldsymbol{P}_{ii}(\tilde{\boldsymbol{\theta}}_i)| < |\boldsymbol{P}_{ii}(\tilde{\boldsymbol{\theta}}_{(i+1)})|. \tag{7}$$

Definition 2 indicates that after learning the second task, the new weights may not be as effective in minimizing the loss of the first task as the weights that we derived by only minimizing the first task. Simply put, our cat-dog binary classifier of the model is no longer working as well as it did prior to learning the rabbit-duck binary classifier.

**Definition 3** (CF-Optimal Class-IL)**:** *A class-IL model $\boldsymbol{\theta}^*$ is called CF-optimal if solely CF is minimized. Specifically, $\boldsymbol{\theta}^*$ is the model minimizing the sum of losses of all diagonal blocks $\boldsymbol{P}_{ii}(\boldsymbol{\theta})$ ignoring the inter-task losses as follows:*

$$\boldsymbol{\theta}^*_{1:T} = \operatorname*{argmin}_{\boldsymbol{\theta}} \sum_{i=1}^{T} |\boldsymbol{P}_{ii}(\boldsymbol{\theta})|. \tag{8}$$

CF-optimal means that the classifier can discriminate between cat-dog (task one); and can discriminate between duck-rabbit (task two), too. Yet, the model does properly classify the classes residing inside each task one and two (intra-task classification), it may still fail at discriminating between different tasks (inter-task classification).

Having defined CF in Definition 2, we will prove its occurrence (in Corollary 1) and its consequent sub-optimality (in Corollary 2), based on Incompatibility Definition and Lemma (Lemma 2). However, before that, it is worth mentioning that in this paper, we make the assumption that tasks are *incompatible*; which is specified in the following. (See Appendix C.)

**Definition 4** (Incompatibility)**:** *Functions $f(x)$ and $g(x)$ which are non-zero are called incompatible and denoted as $f(x) \nparallel g(x)$, if the followings are satisfied:*

$$\frac{df(x)}{dx}\Big|_{x=x^f} = \frac{dg(x)}{dx}\Big|_{x=x^g} = 0, \quad \frac{df(x)}{dx}\Big|_{x=x^g} = \frac{dg(x)}{dx}\Big|_{x=x^f} \neq 0,$$
$$x^f = \operatorname*{argmin}_{x} f(x), \quad x^g = \operatorname*{argmin}_{x} g(x) \quad x^f \neq x^g \tag{9}$$

*where $f(x)$ and $g(x)$ are differentiable at $x^f$ and $x^g$.*

It could be argued that assuming incompatibility of tasks is unfavorable, as good class-IL and task-free algorithms aim to maximize both forward and backward transfer, which would not exist in the case of incompatible tasks. However, the ultimate intent of our paper is to investigate TC and CF in both discriminative and generative modeling settings. Our assumption is designed to capture TC and CF, not forward and backward transfer. While forward and backward transfer are important in class-IL and task-free learning, they are not the focus of our work.

**Lemma 2** (Incompatibility Lemma)**:** *For incompatible $f(x)$ and $g(x)$, i.e., $f(x) \nparallel g(x)$, we can state the following:*

$$x^* \neq x^f, x^g, \quad x^* = \operatorname*{argmin}_{x} f(x) + g(x), \quad x^f = \operatorname*{argmin}_{x} f(x), \quad x^g = \operatorname*{argmin}_{x} g(x). \tag{10}$$

*Proof: See Appendix D.* □

In simple terms, two incompatible tasks (functions) have different minimizers. And, the minimizer of the sum of them is neither of the minimizers of each. This is the case for distinct tasks in practice. From many empirical results [1, 18], we know that always when new tasks are learned the optimal

points of the previous tasks are lost. CF always happens, indicating that incompatibility is always true. With incompatibility we can prove the occurrence of CF in the following corollary.

**Corollary 1** (Catastrophic Forgetting)**:** *For the discriminative class-IL model in Definition 1, due to the sequential diagonal optimization, after optimizing for task $(i+1)$ we can state Eq. 7 implying that CF has occurred for task $i$ when $|\boldsymbol{P}_{ii}(\boldsymbol{\theta})| \nVdash |\boldsymbol{P}_{(i+1)(i+1)}(\boldsymbol{\theta})|$.*

*Proof: See Appendix E.* □

Having proved the occurrence of CF, we also can state, as in the following, that our class-IL model after learning the second task is not even CF-optimal.

**Corollary 2** (Sub-Optimality Corollary)**:** *For the discriminative class-IL model defined in Definition 1, due to the sequential diagonal optimization, after optimizing for task $(i+1)$ the class-IL model may not be CF-optimal if $\sum_{i'=1}^{i} |\boldsymbol{P}_{i'i'}(\boldsymbol{\theta})| \nVdash |\boldsymbol{P}_{(i+1)(i+1)}(\boldsymbol{\theta})|$.*

*Proof: See Appendix E.* □

Based on our mathematical framework, we now define TC (which is a major contribution of this paper) and then the optimal class-IL that aims at minimizing TC and CF (which is the ultimate goal of the class-IL as presented in the very beginning of this paper).

**Definition 5** (Task Confusion)**:** *Consider a class-IL model that minimizes the loss $|\boldsymbol{P}_{ii}(\boldsymbol{\theta})|$ of task $i$, achieving the minimal loss $|\boldsymbol{P}_{ii}(\tilde{\boldsymbol{\theta}}_i)|$, where $\tilde{\boldsymbol{\theta}}_i$ is given by Eq. 6. Then the model proceeds and minimizes the loss $|\boldsymbol{P}_{(i+1)(i+1)}(\boldsymbol{\theta})|$ of task $(i+1)$, achieving the minimal loss $|\boldsymbol{P}_{(i+1)(i+1)}(\tilde{\boldsymbol{\theta}}_{(i+1)})|$. This class-IL model never finds a chance to optimize $\boldsymbol{\theta}$ by minimizing inter-task loss $|\boldsymbol{P}_{(i)(i+1)}|$. Hence, the class-IL model is confused when it comes to distinguishing classes from two distinct tasks because those loss matrices corresponding to inter-task binary classifiers are not minimized jointly.*

Having defined TC, now in the next definition we specify the optimal class-IL model (whose achievement is the ultimate goal of class-IL).

**Definition 6** (Optimal Class-IL)**:** *A class-IL model $\boldsymbol{\theta}^{**}$ is called optimal if both TC and CF are jointly minimized. Specifically, $\boldsymbol{\theta}^{**}$ is the model minimizing the summation of losses of all blocks of the loss matrix including all the inter-task blocks as follows: $\boldsymbol{\theta}^{**}_{1:T,1:T} = \arg\min_{\boldsymbol{\theta}} \sum_{i=1}^{T} \sum_{j=1}^{T} |\boldsymbol{P}_{ij}(\boldsymbol{\theta})|$.*

In the following theorem, we will state our significant discovery that unlike the common belief in the class-IL community, even if CF is prevented, achieving optimal class-IL might be still impossible, and particularly *is* impossible if there is TC due to the failure in minimizing inter-task blocks (non-diagonal blocks) of the loss matrix Eq. 5.

**Theorem 1** (Infeasibility Theorem)**:** *The CF-optimal class-IL model in Definition 3 is not optimal if the entire loss and the diagonal loss are incompatible: $\sum_{i=1}^{T} \sum_{j=1}^{T} |\boldsymbol{P}_{ij}(\boldsymbol{\theta})| \nVdash \sum_{i=1}^{T} |\boldsymbol{P}_{ii}(\boldsymbol{\theta})|$.*

*Proof: See Appendix E.* □

This is interesting because it turns out that achieving optimal class-IL is infeasible even when CF is minimized, due to existence of TC as shown in Fig. 2. In Fig. 2, in the left, we show TC and CF for discriminative class-IL model that is optimized by 'sequen-

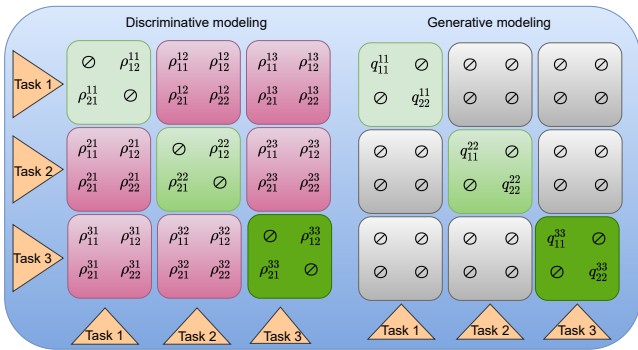

Figure 2: Task Confusion in Discriminative and Generative Modeling.

tially' minimizing the diagonal blocks of the loss matrix. When optimized for the next block, the preceding block loss is gradually forgotten, resulting in CF (lighter green), and, the model is not optimized for inter-task loss matrices, resulting in TC (red). In the right figure, we show CF in generative modeling. When the class-IL model is optimized by 'sequentially' minimizing the diagonal blocks of the loss matrix, the preceding block loss is forgotten when optimized for the next; however, there is no longer any inter-task block (gray).

## 2.2 Generative modeling

In this section, we focus on generative modeling which is promising: it culminates in what we call Feasibility Theorem and offers a solution to address TC. First, we present Joint Probability Equivalence Lemma in the following which helps us to derive the corresponding loss matrix.

**Lemma 3** (Joint Probability Equivalence Lemma): *An $N$-way (class) generative model parameterized by $\boldsymbol{\theta}$, loss function $v(f_{\boldsymbol{\theta}}(x), y)$, and data generating process with probability density $p(x, y)$ is equivalent to the implementation of $N$ distinct generative models with the following loss: $\sum_{i=1}^{N} q_{rr}(\boldsymbol{\theta}) = I_{\boldsymbol{\theta}} = \int_{\mathcal{X} \times \mathcal{Y}} v(f_{\boldsymbol{\theta}}(x), y) p(x, y) \; dx dy$ where $q_{rr}(\boldsymbol{\theta}) = \int_{\mathcal{X}_r \times \mathcal{Y}_r} v(f_{\boldsymbol{\theta}}(x), y) p(x, y) \; dx dy$ in which $\mathcal{X}_r \subset \mathcal{X}, \mathcal{Y}_r \subset \mathcal{Y}, \mathcal{X}_r \cap \mathcal{X}_t = \emptyset, \mathcal{Y}_r \cap \mathcal{Y}_t = \emptyset$, for $r \neq t$. Also, $q_{rr}(\boldsymbol{\theta})$ stands for the loss for the $r$th class.*

A generative class-IL model 'sequentially' trains the model by minimizing the losses of the diagonal blocks of the loss matrix given by

$$
\boldsymbol{Q}(\boldsymbol{\theta}) = \begin{bmatrix} \boldsymbol{Q_{11}}(\boldsymbol{\theta}) & \oslash & \dots & \oslash \\ \oslash & \boldsymbol{Q_{22}}(\boldsymbol{\theta}) & \dots & \oslash \\ \vdots & \vdots & \ddots & \vdots \\ \oslash & \oslash & \dots & \boldsymbol{Q_{TT}}(\boldsymbol{\theta}) \end{bmatrix}, \boldsymbol{Q}_{ii}(\boldsymbol{\theta}) = \begin{bmatrix} q_{11}^{ii}(\boldsymbol{\theta}) & \oslash & \dots & \oslash \\ \oslash & q_{22}^{ii}(\boldsymbol{\theta}) & \dots & \oslash \\ \vdots & \vdots & \ddots & \vdots \\ \oslash & \oslash & \dots & q_{CC}^{ii}(\boldsymbol{\theta}) \end{bmatrix}
$$

(11)

in which $q_{mm}^{ii}$ stands for the loss of the generative model for the $m$th class of task $i$. As we did in the previous section for discriminative modeling in Definitions 1–6, we can define the same properties for generative modeling. In the following theorem, we state that tackling TC through implementation of joint probability $P(X, Y)$ is feasible—via generative modeling. This makes all the difference.

**Theorem 2** (Feasibility Theorem): *For the class-IL model adopting generative modeling with loss matrix in Eq. 11, if CF is prevented, meaning that all diagonal blocks are optimal $[\boldsymbol{Q}_{11}^*(\boldsymbol{\theta}), \boldsymbol{Q}_{22}^*(\boldsymbol{\theta}), \cdots, \boldsymbol{Q}_{NN}^*(\boldsymbol{\theta})]$, the model is optimal.*

*Proof: See Appendix E.* □

In generative modeling, therefore, optimizing for the diagonals is equivalent to optimizing for all the loss terms. In other words, the losses associated with different tasks/classes are irrelevant; therefore, the loss matrix can be only diagonally optimized as shown in Fig. 2. This sums up this section. The lessons learned so far are: (i) class-IL faces two problems, CF and TC, and (ii) TC is inevitable unless we use generative modeling (as proved in Infeasibility/Feasibility Theorems). These are widely applicable lessons for assessing the optimality of the class-IL schemes.

# 3 Optimality analysis of Class-IL strategies

We analyze the behaviors of popular class-IL strategies including (i) regularization, (ii) bias-correction, (iii) replay, and (iv) generative classifier; among which the first three do discriminative modeling, whereas the last one does generative modeling. Note that even generative replay counts as discriminative modeling because eventually a discriminator performs classification. Generative classifier, however, performs classification only/directly via generative modeling. In this section, we study the optimality of the above class-IL strategies. The following three corollaries (i.e., Corollaries 3, 4, and 5) follow Theorem 1; whereas the last one, Corollary 6, follows Theorem 2 (Table 1 summarizes our discussions in this section).

## 3.1 Regularization strategies

We start with regularization. As mentioned, regularization is essentially attempting to preserve the optimality of intra-task blocks (represented by the diagonal blocks $\boldsymbol{P}_{ii}(\boldsymbol{\theta})$'s in Eq. 5) via constraining the proceeding updates to cause as little modifications as possible (e.g., via gradient manipulation), thereby mitigating CF.

**Corollary 3** (Regularization Impotence Corollary): *A regularized class-IL model, which does discriminative modeling, may minimize CF; however, it never achieves optimal class-IL due to sub-optimal TC.*

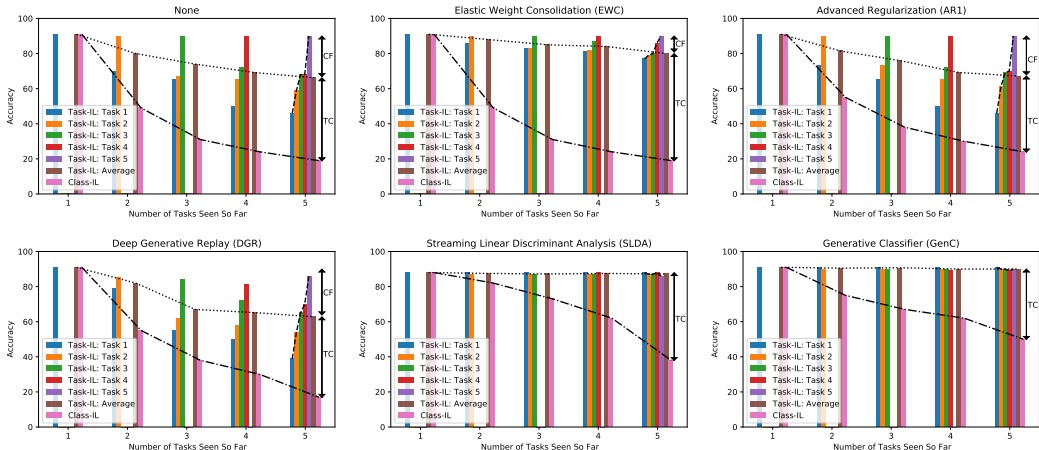

Figure 3: In all the six figures, for the task-IL scenario, the schemes merely face CF (because they are given with the task-ID), and thus, they perform favorably. In the class-IL scenario, however, the models need to discriminate between different tasks, and they usually fail; this is expected due to not minimizing the inter-task block losses.

In class-IL/task-free scenarios, discriminative models' performance can be characterized by a hypothesis. These models, which include regularization schemes like None, LwF, EWC, and SI, prioritize minimizing confusion within tasks over distinguishing between tasks. They achieve this by focusing on optimizing for diagonal loss elements and neglecting off-diagonal elements, which makes them proficient at discriminating classes within tasks but limits their ability to differentiate between tasks. As a result, their classification accuracy is upper-bounded by $100/T\%$, where $T$ is the number of tasks. This limitation suggests that these models are at best CF-optimal, as they prioritize within-task performance over between-task performance, as observed in our experimental results.

**Hypothesis 1** (CF-optimal Model Corollary)**:** *The performance (as measured by classification accuracy) of the CF-optimal class-IL models, which are the None and regularization schemes in Table 2, is upper-bounded by $100/T\%$ where $T$ stands for the number of tasks.*

This can be seen in Table 2 ($T = 5$ for MNIST, CIFAR-10, CORe50 and $T = 10$ for CIFAR-100): when the class-IL schemes only tackle CF not TC such as in the None scheme and the regularization strategy, the performance never exceeds $100/T\%$; because TC (inter-task blocks) is left out sub-optimal[2].

Not only that, the results provided in the work by [4] also support such a *hypothesis*. Nevertheless, there are few results in Masana's [3] works suggesting that regularization schemes augmented with a technique based on entropy demonstrate performances that exceed the $100\%/T$ upper bound although the performance is still far from the performances of schemes based on generative modeling counterparts. This is a slight discrepancy between different bodies of studies: on one hand, the reasoning steps for making such a conclusion seem flawless and there are numerical results to back that up and on the other hand, we cannot ignore the slightly incongruous empirical results. It remains to be investigated how to account for that small increase over the upper bound. This will hopefully be addressed in future works.

So far, we theoretically made it clear what is the distinction between TC and CF; to further empirically distinguish between TC and CF, in Fig. 3 we contrast the performances of the None scheme as well as a typical regularization scheme (Elastic Weight Consolidation, EWC [1] with hyperparameter $\lambda = 5000$) in two scenarios: (i) task-IL, where the model merely faces CF and (ii) class-IL, which the model faces both TC and CF together. The simulations are run with CNN on CIFAR-10 and only the means are reported after 10 repetitions.

As it can be seen in Fig. 3 (top-left) corresponding to the None scheme, TC causes significantly more performance drop than CF. This is because TC has far more block losses ($T^2 - T$ red blocks)

---

[2]Our simulation configuration for the experiments in Table 2 as well as Figs. 3 and 4 follows that of [4]. For more information refer to the README.md file in the code and the supplementary material: Appendices F to H.

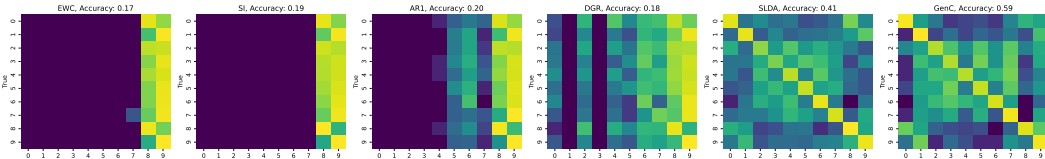

Figure 4: Generative classifiers like SLDA and GenC mitigate TC and CF (CIFAR-10).

Table 1: Comparison of different strategies in addressing CF, TC, and Bias-Correction (BC).

| Strategies | CF | TC | BC | Theoretical Remarks |
|---|---|---|---|---|
| Regularization | ✓ | ✗ | ✗ | see Corollary 3 (Regularization Impotence Corollary) and Hypothesis 1 (CF-optimal Model Corollary) |
| Distillation | ✓ | ✗ | ✓ | since distillation is essentially regularization Corollary 3 (Regularization Impotence Corollary) and Hypothesis 1 (CF-optimal Model Corollary) |
| Bias-correction | ✗ | ✗ | ✓ | see Corollary 4 (Bias-Correction Impotence Corollary) |
| Generative replay | ✓ | ✓ | ✓ | see Corollary 5 (Generative Replay Corollary) |
| Generative classifier | ✓ | ✓ | ✓ | see Corollary 6 (Generative Classifier Feasibility Corollary) |

than CF ($T$ green blocks) to be minimized as it can be seen in Fig. 2 (left). In Fig. 3 (top-middle) corresponding to the EWC scheme, we observe that although EWC is able to minimize CF, the amount of TC remains unchanged indicating the ineffectiveness of regularization for TC (see Appendix F).

## 3.2 Distillation strategies

Knowledge distillation [19] serves as an effective regularization strategy in mitigating catastrophic forgetting. Unlike approaches such as EWC and SI, which impose constraints on parameter updates, knowledge distillation focuses on ensuring consistency in the responses of the new and old models. This distinctive feature provides a broader solution space, enabling the model to explore optimal parameters that cater to both new and old tasks.

However, because the knowledge distillation strategy inherently operates as a regularization technique, it is upper-bounded like all other regularization strategies, as outlined in Hypothesis 1. This upper-boundedness can be observed in Table 2 for *Dis* scheme [19].

## 3.3 Bias-correction strategies

Bias-correction specifically attempts to mitigate TC; however, only minutely (see Fig. 3 (top-right) and Fig. 4 pertaining to the AR1 scheme): it removes the bias from inter-task binary classifiers, slightly reducing the inter-task block losses. Nonetheless, bias-correction is not enough to achieve optimal inter-task discrimination, since bias parameters constitute a fringe minority of all the parameters of models.

**Corollary 4** (Bias-Correction Impotence Corollary)**:** *For a bias-corrected class-IL model, which does discriminative modeling, neither the optimality of diagonal blocks $\boldsymbol{P}_{ii}(\boldsymbol{\theta})$ is ensured nor inter-task blocks $\boldsymbol{P}_{ij}(\boldsymbol{\theta})$ for $i \neq j$; therefore it never achieves optimal class-IL.*

The slight improvement in CWR [20], CWR+, AR1 [21], and Label [11], via correcting the biases shows itself in Table 2, where the bias-corrected schemes outperform the schemes of regularization (and None) by doing as little as only correcting the biases, thereby mitigating TC. This suggests the priority of the TC problem over CF.

## 3.4 Generative replay

On the other hand, there exists the generative replay strategy that attempts to minimize the objective function in Eq. 1 via a surrogate density $\hat{p}(x, y)$, which generates pseudo samples, to mimic the real density of $p(x, y)$. For generative replay we can present the following corollary.

**Corollary 5** (Generative Replay Corollary)**:** *A generative replay-based class-IL model, which is discriminative, possessing a surrogate density $\hat{p}(x, y)$ can achieve optimal class-IL iff $\hat{p}(x, y)$ is identical to the real density $p(x, y)$.*

Although in principle generative replay can achieve optimality and cope with TC and CF because when learning each new task all inter-task and intra-task blocks are minimized, in practice, it is challenging to efficiently train such a surrogate density $\hat{p}(x, y)$, where $\hat{p}(x, y) \sim p(x, y)$ (unless by explicitly storing all/exemplars of the dataset); this can be seen in Table 2 where the family of generative replay, DGR [6], BI-R, and BI-R+SI [18], although does well for toy datasets (MNIST), it fails for larger datasets in competition with generative classifier. This can also be seen in Fig. 3 (bottom-left) and Fig. 4 where DGR not only fails to cope with TC but also suffers from CF due to possessing a poor surrogate density $\hat{p}(x, y)$ which cannot capture the real density $p(x, y)$.

### 3.5 Generative classifier

Eventually, unlike the previous three strategies (all discriminative modeling), the fourth strategy, the generative classifier (relying on generative modeling), represented by SLDA [10] and GenC [4] not only can dispense with rehearsal without worrying about TC, but also promises optimal class-IL as presented in the following corollary.

**Corollary 6** (Generative Classifier Feasibility Corollary)**:** *Following Feasibility Theorem, for a generative class-IL model, which does generative modeling, when CF is minimized, the class-IL model is optimal.*

We see in Table 2 (and Figs. 3 (bottom-middle) and (bottom-right)) that SLDA [10] and GenC [4] can best cope with TC. For preventing CF,

Table 2: The means and $\pm$ SEMs of accuracies in class-IL scenarios with 10 runs on four benchmarks.

| Scheme | MNIST | CIFAR-10 | CIFAR-100 | CORe50 |
|---|---|---|---|---|
| \multicolumn{5}{c}{Lower Bound (None) and Upper Bound (Joint)} | | | | |
| None | 19.92±0.02 | 18.74±0.29 | 7.96±0.11 | 18.65±0.26 |
| Joint | 98.23±0.04 | 82.07±0.15 | 54.08±0.27 | 71.85±0.30 |
| \multicolumn{5}{c}{Regularization Strategy} | | | | |
| EWC | 19.93±0.06 | 18.77±0.31 | 8.41±0.09 | 18.70±0.27 |
| SI | 19.88±0.09 | 18.00±0.33 | 9.32±0.07 | 18.61±0.22 |
| \multicolumn{5}{c}{Distillation Strategy} | | | | |
| Dis | 19.87±0.08 | 18.31±0.44 | 9.79±0.13 | 19.35±0.31 |
| \multicolumn{5}{c}{Bias-Correction Strategy} | | | | |
| CWR | 30.96±2.33 | 18.63±1.44 | 21.98±0.57 | 40.11±1.15 |
| CWR+ | 39.02±2.88 | 22.69±1.17 | 9.29±0.19 | 40.78±1.05 |
| AR1 | 49.38±2.36 | 25.13±1.18 | 21.01±0.51 | 44.13±1.06 |
| Label | 33.01±2.01 | 19.21±1.22 | 23.35±0.31 | 41.55±1.01 |
| \multicolumn{5}{c}{Generative Replay Strategy} | | | | |
| DGR | 91.12±0.65 | 18.13±1.85 | 9.41±0.30 | - |
| BI-R | - | - | 21.41±0.19 | 61.04±1.01 |
| BI-R+SI | - | - | 34.34±0.23 | 62.51±0.29 |
| \multicolumn{5}{c}{Generative Classifier Strategy} | | | | |
| SLDA | 87.31±0.02 | 38.33±0.04 | 44.49±0.00 | 70.80±0.00 |
| GenC | 93.75±0.09 | 56.02±0.04 | 49.53±0.07 | 70.80±0.10 |

however, these two schemes follow a shared approach which is adopting expansion-based architecture: instead of using the same architecture for all classes, and therefore, forgetting previous classes when new classes are learned, expansion-based architectures grow their model as new classes are learned. Therefore, they do not overwrite new knowledge on the previous knowledge. This is why schemes like SLDA [10] and GenC [4] suffer from no CF in Figs. 3 (bottom-middle) and (bottom-right). Also, this can be seen in Fig. 4.

## 4 Big picture and conclusion

Since the advent of AlexNet [22], the neuroscience community has been skeptical of the deep learning community's discriminative modeling approach for classification [23, 24]. They argue that humans do not learn $p(y|x)$ for classification (discriminative modeling); instead, humans learn $p(x)$ which is generative modeling. The primary issue with discriminative modeling is shortcut learning [24], a concern that has recently gained more attention within the deep learning community [25]. In this work, we investigate how shortcut learning can particularly hinder class-incremental learning. We discuss how shortcut learning in discriminative modeling leads to task confusion and argue that generative modeling, in principle, addresses this issue.

We proposed a mathematical framework to formalize problems of class-incremental learning and task-free: task confusion and catastrophic forgetting. We proved that in discriminative modeling the non-diagonal block losses are not minimized, which causes task confusion resulting in sub-optimal performance for the class-incremental learning model even though catastrophic forgetting is prevented. We presented our empirical results confirming that generative modeling does not suffer from task confusion because there are no non-diagonal blocks that need to be minimized. We observed that while generative modeling is effective for coping with task confusion, adopting expansion-based architectures can overcome catastrophic forgetting.

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

# A  Complexity analysis

## Appendix A

### Conditional Probability Equivalence Lemma (Lemma 1)

The loss function of an $N$-way classifier is mathematically equivalent to the combined loss functions of $\binom{N}{2}$ binary classifiers. This approach ensures that we account for each pair of classes only once, thereby avoiding any overlap. Starting with a simple scenario where $N = 2$, the loss function is defined as:

$$I_{\boldsymbol{\theta}} = \int_{\mathcal{X} \times \{1,2\}} v(f_{\boldsymbol{\theta}}(x), y) p(x, y) \, dx \, dy. \tag{12}$$

This setup is naturally a binary classifier, focusing solely on one class pair. When considering $N = k$, with $k \geq 2$, we assume the loss function can be expanded to:

$$I_{\boldsymbol{\theta}} = \frac{1}{k-1} \sum_{i=1}^{k} \sum_{j=i+1}^{k} \int_{\mathcal{X} \times \{i,j\}} v(f_{\boldsymbol{\theta}}(x), y) p(x, y) \, dx \, dy. \tag{13}$$

Here, each class combination $(i, j)$, where $i < j$, is included once, optimizing the calculation and removing redundancy. For example, when $N = 3$, we break it down as follows:

$$I_{\boldsymbol{\theta}} = \frac{1}{2} \left( \int_{\mathcal{X} \times \{1,2\}} v(f_{\boldsymbol{\theta}}(x), y) p(x, y) \, dx \, dy + \int_{\mathcal{X} \times \{1,3\}} v(f_{\boldsymbol{\theta}}(x), y) p(x, y) \, dx \, dy \tag{14}$$

$$+ \int_{\mathcal{X} \times \{2,3\}} v(f_{\boldsymbol{\theta}}(x), y) p(x, y) \, dx \, dy \right). \tag{15}$$

Introducing an additional class, $k + 1$, results in more binary comparisons:

$$I_{\boldsymbol{\theta}} = \frac{1}{k} \left( \sum_{i=1}^{k} \sum_{j=i+1}^{k} \int_{\mathcal{X} \times \{i,j\}} v(f_{\boldsymbol{\theta}}(x), y) p(x, y) \, dx \, dy + \sum_{i=1}^{k} \int_{\mathcal{X} \times \{i,k+1\}} v(f_{\boldsymbol{\theta}}(x), y) p(x, y) \, dx \, dy \right). \tag{16}$$

Conclusively, for any $N$:

$$I_{\boldsymbol{\theta}} = \frac{1}{N-1} \sum_{i=1}^{N} \sum_{j=i+1}^{N} \int_{\mathcal{X} \times \{i,j\}} v(f_{\boldsymbol{\theta}}(x), y) p(x, y) \, dx \, dy. \tag{17}$$

This equation confirms that the loss function for an $N$-way classifier replicates that of several binary classifiers, with each class pair uniquely represented.

## Appendix B

Note that in Eq. 5, we applied the result that we derived from Lemma 1 to the class-IL scenario; however, we modified it with new notations considering the context and nuances of class-IL. The result is that the loss is now atomized in a hierarchical manner: the ultimate loss, the task-level loss, and finally each binary classifier's loss. The ultimate loss is the summation of a combination of task-level losses that are matrices; the loss of each task is in turn a matrix of losses that are to be summed, to account for all the binary classifiers inside each task for all classes.

## Appendix C

*The Incompatibility Assumption and Why It Is a Realistic Assumption.* The incompatibility assumption is realistic due to the following reasons: (i) in class-IL systems, different tasks have their own modalities due to having unique data distributions and therefore they have their own minimizers. (ii) At the minimizer of one task where its gradient is zero, the other tasks have non-zero gradients.

This is tenable from what we know concerning the empirical results in the literature [18] because if the other tasks' gradients were also zero, then when learning the other tasks, there would not be any changes in weights; however, we know that new tasks change the weights as soon as they arrive. Hence, they must have non-zero gradients. Both these assumptions are realistic and observed in many works [1]. Simply put, if two distinct tasks did not have different minimizers, there would not be any CF in practice in the first place.

## Appendix D

*Explanation and Proof of the Incompatibility Lemma (Lemma 2).* In simple terms, two incompatible tasks (functions) have different minimizers. And, the minimizer of the sum of them is neither of the minimizers of each. This is the case for distinct tasks in practice. From many empirical results [1, 18], we know that always when new tasks are learned the optimal point of the previous tasks are lost. CF always happens, indicating that incompatibility is always true.

### Incompatibility Lemma (Lemma 2)

For Incompatibility Lemma, which addresses the relationship between the minimizers of two incompatible functions $f(x)$ and $g(x)$, denoted as $f(x) \nmid g(x)$. This lemma posits that the minimizer of the sum $f(x) + g(x)$ is distinct from the minimizers of $f(x)$ and $g(x)$ individually, indicated by:

$$x^* \neq x^f, x^g, \quad x^* = \arg\min_x f(x) + g(x), \quad x^f = \arg\min_x f(x), \quad x^g = \arg\min_x g(x). \quad (18)$$

To begin, we define two functions as incompatible if the minimization of one does not necessarily imply the minimization of the other. In other words, their minimizers do not coincide. Suppose, for contradiction, that the minimizer $x^f$ of $f(x)$, is also the minimizer of the combined function $f(x) + g(x)$. This would imply:

$$x^f = \arg\min_x f(x) = \arg\min_x \left( f(x) + g(x) \right). \quad (19)$$

Given that at $x^f$, the derivative of $f(x)$ with respect to $x$ must equal zero, we also assume:

$$\left. \frac{df(x)}{dx} \right|_{x=x^f} = 0. \quad (20)$$

If $x^f$ were also a minimizer of $f(x) + g(x)$, the derivative of the sum at $x^f$ would similarly vanish:

$$\left. \frac{d}{dx} \left( f(x) + g(x) \right) \right|_{x=x^f} = 0. \quad (21)$$

This would suggest that the derivative of $g(x)$ at $x^f$ must also equal zero, leading to:

$$\left. \frac{dg(x)}{dx} \right|_{x=x^f} = 0. \quad (22)$$

However, given that $f(x)$ and $g(x)$ are incompatible, $\left. \frac{dg(x)}{dx} \right|_{x=x^f} \neq 0$, indicating that there exists a direction opposite to the gradient of $g(x)$ at $x^f$ which can further decrease $f(x) + g(x)$. This contradiction shows that $x^f$ cannot be the minimizer of $f(x) + g(x)$.

Applying a symmetric argument for $x^g$, suppose $x^g$ is both the minimizer of $g(x)$ and $f(x) + g(x)$:

$$x^g = \arg\min_x g(x) = \arg\min_x \left( f(x) + g(x) \right). \quad (23)$$

Following the same logic as before, we derive that the derivative of $f(x)$ at $x^g$ should be zero, leading to:

$$\left. \frac{df(x)}{dx} \right|_{x=x^g} = 0. \quad (24)$$

Yet, since the functions are incompatible, $\frac{df(x)}{dx}\Big|_{x=x^g} \neq 0$, reinforcing our contradiction and showing that $x^g$ is not the minimizer of the combined function either.

In conclusion, $x^*$, the minimizer of $f(x) + g(x)$, is distinct from both $x^f$ and $x^g$, illustrating that when functions $f(x)$ and $g(x)$ are incompatible, their individual minimizers cannot optimize the combined function.

## Appendix E

**Catastrophic Forgetting Corollary (Corollary 1)**

To address the phenomenon of Catastrophic Forgetting (CF) in discriminative class-incremental learning (class-IL) models, we must consider how sequential optimization impacts task performance. When a class-IL model trains on a specific task $i$, it aims to optimize parameters $\boldsymbol{\theta}_i$ to achieve minimal loss for that task:

$$\tilde{\boldsymbol{\theta}}_i = \operatorname*{argmin}_{\boldsymbol{\theta}} |\boldsymbol{P}_{ii}(\boldsymbol{\theta})|, \tag{25}$$

ensuring optimal performance for task $i$.

As the model proceeds to train on a subsequent task $i + 1$, it again seeks to optimize its parameters, but this time for the new task-specific loss:

$$\tilde{\boldsymbol{\theta}}_{(i+1)} = \operatorname*{argmin}_{\boldsymbol{\theta}} |\boldsymbol{P}_{(i+1)(i+1)}(\boldsymbol{\theta})|. \tag{26}$$

This optimization results in a new set of parameters $\tilde{\boldsymbol{\theta}}_{(i+1)}$, ideally suited to minimize the loss for task $i + 1$. However, these two tasks are incompatible:

$$|\boldsymbol{P}_{ii}(\boldsymbol{\theta})| \nparallel |\boldsymbol{P}_{(i+1)(i+1)}(\boldsymbol{\theta})|, \tag{27}$$

indicating that the optimal parameters for task $i + 1$ do not coincide with those for task $i$. This misalignment implies that the gradient of the loss function for task $i$ evaluated at the parameters optimized for task $i + 1$ is non-zero:

$$\bigtriangledown_{\boldsymbol{\theta}} |\boldsymbol{P}_{ii}(\boldsymbol{\theta})|\big|_{\boldsymbol{\theta}=\tilde{\boldsymbol{\theta}}_{(i+1)}} \neq 0. \tag{28}$$

This non-zero gradient underscores that the parameters $\tilde{\boldsymbol{\theta}}_{(i+1)}$ are not at a local minimum for task $i$, revealing that there are still directions in parameter space that could reduce the loss for task $i$ if adjustments were made.

Consequently, the application of these parameters to task $i$ results in increased loss:

$$|\boldsymbol{P}_{ii}(\tilde{\boldsymbol{\theta}}_i)| < |\boldsymbol{P}_{ii}(\tilde{\boldsymbol{\theta}}_{(i+1)})|. \tag{29}$$

This increase in loss is a direct manifestation of Catastrophic Forgetting, highlighting that performance on task $i$ deteriorates as the model is sequentially optimized for task $i + 1$.

**Sub-Optimality Corollary (Corollary 2)**

To address the sub-optimality issue in discriminative class-incremental learning (class-IL) models due to sequential optimization, we need to consider how the model is optimized across multiple tasks. For tasks up to task $i$, the optimization can be collectively represented as

$$\sum_{i'=1}^{i} |\boldsymbol{P}_{i'i'}(\boldsymbol{\theta})|, \tag{30}$$

where $\boldsymbol{\theta}$ reflects the parameters that have been adapted and optimized through task $i$.

As the training progresses to the next task, $i + 1$, the parameters are further optimized to minimize the loss specific to this new task, represented by

$$\tilde{\boldsymbol{\theta}}_{(i+1)} = \operatorname*{argmin}_{\boldsymbol{\theta}} |\boldsymbol{P}_{(i+1)(i+1)}(\boldsymbol{\theta})|. \tag{31}$$

This optimization yields a new set of parameters ideally suited for minimizing the loss for task $i + 1$.

However, the inherent incompatibility of the optimization objectives for the accumulated previous tasks and the new task becomes apparent through the expression

$$\sum_{i'=1}^{i} |\boldsymbol{P}_{i'i'}(\boldsymbol{\theta})| \nparallel |\boldsymbol{P}_{(i+1)(i+1)}(\boldsymbol{\theta})|. \tag{32}$$

This notation indicates a fundamental misalignment between the cumulative optimization goals for earlier tasks and the optimization for the new task. It implies that the parameters optimized for task $i + 1$ do not coincide with minimizing the cumulative loss of the earlier tasks.

When these parameters $\tilde{\boldsymbol{\theta}}_{(i+1)}$ are applied to evaluate the cumulative loss from previous tasks, the gradient of this cumulative loss function evaluated at the new parameters is not minimized, as shown by

$$\triangledown_{\boldsymbol{\theta}} \left( \sum_{i'=1}^{i} |\boldsymbol{P}_{i'i'}(\boldsymbol{\theta})| \right) \Big|_{\boldsymbol{\theta}=\tilde{\boldsymbol{\theta}}_{(i+1)}} \neq 0. \tag{33}$$

This non-zero gradient indicates that there are still directions in which the parameters can be adjusted to further reduce the loss for previous tasks, underscoring that $\tilde{\boldsymbol{\theta}}_{(i+1)}$ is not at a minimum for the accumulated task losses.

The practical implication of this analysis is that using the parameters optimized for task $i + 1$ on the cumulative tasks results in an increase in the overall loss for tasks 1 through $i$:

$$\sum_{i'=1}^{i} |\boldsymbol{P}_{i'i'}(\tilde{\boldsymbol{\theta}}_i)| < \sum_{i'=1}^{i} |\boldsymbol{P}_{i'i'}(\tilde{\boldsymbol{\theta}}_{(i+1)})|.$$

This increase in cumulative loss for the earlier tasks when parameters are optimized for a subsequent task underlines the sub-optimal performance of the model for those earlier tasks. This finding highlights the necessity for strategies that can effectively manage the incompatibility between tasks, such as those involving holistic optimization approaches that consider all tasks simultaneously or mechanisms that introduce less forgetful learning dynamics to ensure more uniform performance across all tasks.

**Infeasibility Theorem (Theorem 1)**

To formally establish that the CF-optimal class-incremental learning (class-IL) model does not achieve optimal performance when the overall loss and the diagonal loss are incompatible, we need to delve into the distinctions and relationships between these two loss frameworks. The total loss for a class-IL model accounts for all interactions between class pairs across $T$ tasks, capturing both intra-task (diagonal elements) and inter-task (off-diagonal elements) dynamics. Mathematically, this can be represented as:

$$\text{Total Loss} = \sum_{i=1}^{T} \sum_{j=1}^{T} |\boldsymbol{P}_{ij}(\boldsymbol{\theta})|, \tag{34}$$

which sums the losses for all class pairings, reflecting the complexity and interconnectedness of all tasks.

Conversely, the diagonal loss focuses solely on the intra-task interactions, optimizing to minimize the forgetting of previously learned tasks as new ones are introduced. This is quantified by:

$$\text{Diagonal Loss} = \sum_{i=1}^{T} |\boldsymbol{P}_{ii}(\boldsymbol{\theta})|, \tag{35}$$

where the loss is accumulated solely from the diagonal components corresponding to each task-specific loss.

The proof's critical aspect is demonstrating the incompatibility between these two optimization targets, which is highlighted when we find that the minimization of the diagonal loss does not necessarily coincide with the minimization of the total loss:

$$\sum_{i=1}^{T} \sum_{j=1}^{T} |\boldsymbol{P}_{ij}(\boldsymbol{\theta})| \nparallel \sum_{i=1}^{T} |\boldsymbol{P}_{ii}(\boldsymbol{\theta})|. \tag{36}$$

This expression $\nparallel$ explicitly denotes that the optimal solutions for these two objectives do not align.

If we assume that the CF-optimal parameters, $\tilde{\boldsymbol{\theta}}_{CF}$, are those that have been tuned to specifically minimize the diagonal loss:

$$\tilde{\boldsymbol{\theta}}_{CF} = \operatorname*{argmin}_{\boldsymbol{\theta}} \sum_{i=1}^{T} |\boldsymbol{P}_{ii}(\boldsymbol{\theta})|, \tag{37}$$

and then apply these parameters to evaluate the total loss, they do not minimize it, indicating a misalignment:

$$\tilde{\boldsymbol{\theta}}_{CF} \neq \operatorname*{argmin}_{\boldsymbol{\theta}} \sum_{i=1}^{T} \sum_{j=1}^{T} |\boldsymbol{P}_{ij}(\boldsymbol{\theta})|. \tag{38}$$

This leads us to the observation that because these optimization goals are not aligned, the gradient of the total loss evaluated at $\tilde{\boldsymbol{\theta}}_{CF}$ is non-zero:

$$\nabla_{\boldsymbol{\theta}} \left( \sum_{i=1}^{T} \sum_{j=1}^{T} |\boldsymbol{P}_{ij}(\boldsymbol{\theta})| \right) \Bigg|_{\boldsymbol{\theta} = \tilde{\boldsymbol{\theta}}_{CF}} \neq 0. \tag{39}$$

Such a gradient indicates that the parameters that are optimal for minimizing the diagonal loss do not provide a minimum for the total loss landscape, confirming the sub-optimality of the CF-optimal model when considering the broader spectrum of tasks. This illustrates the critical need for optimization strategies in class-IL models that take into account not just the task-specific losses but also the interdependencies between different tasks to achieve truly optimal performance.

**Joint Probability Equivalence Lemma (Lemma 3)**

To rigorously establish that an $N$-way generative model operating on the entire dataset is equivalent to operating $N$ separate class-specific models, we start by defining the model and its operational components. The overall model applies a loss function $v(f_{\boldsymbol{\theta}}(x), y)$ across the joint data set $\mathcal{X} \times \mathcal{Y}$, parameterized by $\boldsymbol{\theta}$. This setup calculates the total loss as:

$$I_{\boldsymbol{\theta}} = \int_{\mathcal{X} \times \mathcal{Y}} v(f_{\boldsymbol{\theta}}(x), y) p(x, y) \, dx \, dy, \tag{40}$$

encompassing all interactions within the dataset. In contrast, we define $N$ distinct generative models, each tailored to a specific class. These models focus on subsets $\mathcal{X}_r$ and $\mathcal{Y}_r$ that are exclusive to each class, ensuring no overlap in data across these models. The loss for each class-specific model is given by:

$$q_{rr}(\boldsymbol{\theta}) = \int_{\mathcal{X}_r \times \mathcal{Y}_r} v(f_{\boldsymbol{\theta}}(x), y) p(x, y) \, dx \, dy, \tag{41}$$

where $q_{rr}(\boldsymbol{\theta})$ calculates the loss within the confines of each class's data subset.

The crux of the proof lies in demonstrating that the sum of the losses from each individual class-specific model is equivalent to the total loss computed across the entire dataset. Given that the subsets $\mathcal{X}_r$ and $\mathcal{Y}_r$ are mutually exclusive and collectively exhaustive, the union of all these subsets covers the full data space $\mathcal{X} \times \mathcal{Y}$. Hence, the aggregate of all class-specific losses matches the overall model's loss:

$$\sum_{r=1}^{N} q_{rr}(\boldsymbol{\theta}) = \sum_{r=1}^{N} \int_{\mathcal{X}_r \times \mathcal{Y}_r} v(f_{\boldsymbol{\theta}}(x), y) p(x, y) \, dx \, dy = \int_{\mathcal{X} \times \mathcal{Y}} v(f_{\boldsymbol{\theta}}(x), y) p(x, y) \, dx \, dy = I_{\boldsymbol{\theta}}. \tag{42}$$

This equivalence solidifies the understanding that dividing a complex, multi-class generative model into simpler, class-specific models does not compromise the integrity of loss computation.

**Feasibility Theorem (Theorem 2)**

In a generative class-incremental learning (class-IL) model, the structure of the loss matrix $\boldsymbol{Q}(\boldsymbol{\theta})$ is such that it isolates each task's performance in its own diagonal block, $\boldsymbol{Q}_{ii}(\boldsymbol{\theta})$, and eliminates inter-task loss contributions through null off-diagonal blocks. Here, CF optimality is not merely about achieving minimal loss for each individual task, but rather about reducing the sum of these diagonal losses to the lowest possible level across the entire model:

$$\min_{\boldsymbol{\theta}} \sum_{i=1}^{N} \boldsymbol{Q}_{ii}(\boldsymbol{\theta}). \tag{43}$$

This ensures that all tasks are optimized simultaneously under a unified parameter set, $\boldsymbol{\theta}$.

The proof of the system's overall optimality, then, is directly tied to this approach. Since the off-diagonal elements are zero—indicating no cross-task interference—the total loss of the model simplifies to the sum of its diagonal blocks:

$$\sum_{i=1}^{N} \sum_{j=1}^{N} \boldsymbol{Q}_{ij}(\boldsymbol{\theta}) = \sum_{i=1}^{N} \boldsymbol{Q}_{ii}(\boldsymbol{\theta}). \tag{44}$$

This relationship underscores that minimizing the diagonal is essentially minimizing the entire matrix.

By achieving CF optimality—where the total diagonal loss is minimized—the model not only secures the lowest possible loss for each task but also assures that learning new tasks doesn't degrade the performance on any of the previously learned tasks. This makes the entire generative class-IL model optimal:

$$\boldsymbol{\theta}^* = \operatorname*{argmin}_{\boldsymbol{\theta}} \sum_{i=1}^{N} \boldsymbol{Q}_{ii}(\boldsymbol{\theta}) = \operatorname*{argmin}_{\boldsymbol{\theta}} \sum_{i=1}^{N} \sum_{j=1}^{N} \boldsymbol{Q}_{ij}(\boldsymbol{\theta}). \tag{45}$$

## Appendix F

*Comprehensive Explanations of Simulation Results for Class-IL and Task-IL.* Table 2 provided only the final brief results of performances for different schemes merely in the class-IL scenario. However, in order to fully appreciate the challenge of class-IL, it is imperative to distinguish between TC and CF; this entails to contrast the performances of schemes in class-IL and task-IL regimes. This comparison reveals an important distinction: even if CF is somehow tackled, TC is still the bottleneck in class-IL.

Indeed, the schemes delivering the best results not only solve CF but also are the most effective at tackling TC. We present our results only for CIFAR-10 [26] and these results with their consequent insights carry over to other datasets; therefore, we refrain from reporting figures pertaining to all other datasets.

In Fig. 3 (top-left) no scheme is used to tackle TC or CF; here this is denoted by *None* consistent with the notation used in Table 2. In this figure, we progressively demonstrate the performances of the model on task one, two, three, four, and five shown with blue, orange, green, red, and purple, respectively. The brown color stands for the average of the aforementioned task performances (the average task-IL performance).

Meanwhile, the pink color is used for the performance of the model over all the tasks that it has been trained so far without providing the task-ID (the class-IL performance). When the model learns the first task the values of task-IL and class-IL performance are the same. However, they diverge as new tasks are learned.

Observing the deterioration of each individual task's accuracy as well as the average accuracy in task-IL scenario in Fig. 3 (top-left), it can be seen that this case suffers from CF: learning new tasks causes old ones to be forgotten. However, when TC is considered, which is when we switch to the class-IL scenario, the deterioration is much worse because now not only the model suffers from forgetting but also it is confused among new tasks when they are learned.

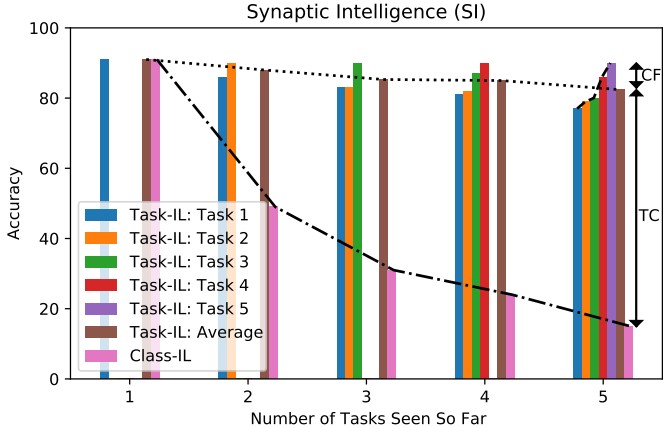

Figure F.1: In this figure, Synaptic Intelligence (SI) [2] with $\lambda = 1$ is adopted. It is clear that SI is almost effective at mitigating CF; however, ineffective for TC.

SI [2] is surprisingly successful at tackling CF and unsurprisingly ineffective for TC. The latter surprise is because as it can be seen in Fig. F.1 the accuracy drop inflicted by CF is considerably mitigated. This is surprising because in the regularization strategy the model has to update the parameters of the previous tasks to learn a new task; hence, forgetting intuitively seems inevitable given the constant weight overwriting; thus, ameliorating forgetting as much as it is achieved by SI deserves admiration (it is surprising).

Nonetheless, the model still unsurprisingly suffers from severe confusion and seemingly the regularization had no impact on tackling TC at all, which is what we predicted in the Regularization Impotence Corollary. Because regularization does absolutely nothing to remedy the problem of sub-optimal inter-task blocks. Ergo, the ineffectiveness is totally predicted and unsurprising.

AR1 [21] by correcting the biases slightly enhances the accuracy of the class-IL scenario (see Fig. 3 (top-right)); it may slightly mitigate TC. However, the model still suffers from TC and CF.

SLDA [10] in Fig. 3 (bottom-middle) is another exemplar scheme of the generative classifier strategy similar to [4]. Again, we observe what is expected thanks to our theoretical framework: generative classifier has no problem with TC. Also, we see that via model isolation SLDA manages to tackle CF.

## Appendix G

*Source Code Availability and Reproducibility.* To produce our results, we used the code attached. We made some adjustments to suit the code for our purposes, mostly for visualizations. Our data is available. Also, the code to generate the data which was used to generate all the figures and tables are available in the supplementary material. All the code pertaining to the visualizations (figures) are also available.

As mentioned above, we adopted and adapted the source code of the work in [4]. The reason for choosing this repository among all other alternatives is the success of this repository in ensuring a standardized training regime across many popular class-incremental learning algorithms. Because one of the critical issues in the literature on class-incremental learning (which reflects itself in the source codes) is that there are many different regimes with which schemes are evaluated. This makes a fair comparison impossible. For example, as we mentioned in the main text, there are two categories of incremental learning: task-based and task-free:

The task-based strategies assume the task-ID is present at training and/or test while task-free strategies banish the notion of task. These two strategies have different simulation regimes and are hard to compare. The task-based strategy itself consists of three scenarios: task-incremental, domain-incremental, and class-incremental. All with their own simulation regimes.

Then in how the benchmarks are defined, there are differences in the literature: for example, some strategies assume that the training dataset for each task is available all at once whereas others prefer to feed the neural networks with a stream of data. As a result, it is often hard to make fair comparisons.

For example, Elastic Weight Consolidation (EWC) [1] assumes that when learning each task, besides the task-ID, the entire task is available at once for the learner to look at; whereas Synaptic Intelligence (SI) [2], Deep Generative Replay [6], BI-R, BI-R+SI [18], and Labels Trick [11] take data in a stream fashion although they still suppose the task-ID is given like EWC.

There is another category of works who imposes a more restricted training regime that entirely does away with Task-ID; not only that, this category also only reveals the data to the learner in a stream fashion. Schemes like CWR, CWR+ [20], AR1 [21], SLDA [10], and Generative Classifier [4] are of this sort.

The chosen code repository always stays clear on all these differences in order to let the reader make a more informed comparison/judgment.

## Appendix H

*Hyperparameter Specification.* We run our experiments with the same hyperparameters as in [4] for consistency as well as reproducibility of the previous findings. In our experiments concerning the MNIST benchmark the 10 classes are sorted out in 5 tasks where each task contains two classes; for example, the first task has digits zero and one. And so on it goes for the second, third, fourth, and fifth task. CIFAR-10 and CORe50 with 10 classes also has two classes in each task. In contrast, CIFAR-100 offers ten tasks with each presenting ten classes.

Concerning the training regime of the MNIST, CIFAR-10, CIFAR-100, and CORe50 benchmarks, we ran the experiments for 2000, 5000, 5000, *single-pass* iterations, respectively, with mini-batch sizes of 128, 256, 256, and 1, via the learning rates of 0.001, 0.001, 0.001, and 0.0001. Following the simulation scenario of [4] for the CIFAR-100 dataset there is a pretrained model being used which is trained on CIFAR-10 as also specified, programmed, and open-sourced in [18]. The pretrained model is a modified version of the ResNet18 architecture. For CORe50, however, the pretrained model is trained on ImageNet.

Different incremental learning schemes of course come with their own hyperparameters; and being specific about the choice of hyperparameters is essential for upholding the scientific explicity and integrity. In the following, we therefore explicate the hyperparameters used for each of the schemes in our experiments:

EWC uses the values of $10^6$, 10, 100, and 10 for MNIST, CIFAR-10, CIFAR-100, and CORe50 respectively as the values of $\lambda$ after grid-searching over

$$[10^{-1}, 10^{-2}, \cdots, 10^7].$$

Meanwhile, SI for $\lambda$ uses $10^3$, 1, 1, and 10 for four datasets after searching over

$$[10^{-3}, 10^{-2}, \cdots, 10^9].$$

Following the simulation convention in [4] via grid search in $[0, 10, \cdots, 90]$ the parameter $X$ for BI-R is selected which is 70 and 0 for CIFAR-100 and CORe50, respectively. While concerning BI-R+SI scheme for $X$ and $\lambda$ in $[0, 20, \cdots, 90]$ and $[10^{-3}, 10^{-2}, \cdots, 10^9]$ values of $10^8$ and 0.01 were chosen.

AR1 [21] uses 10, 100, $10^3$, and 1 for four datasets after grid-searching over $[10^{-3}, 10^{-2}, \cdots, 10^9]$ for $\lambda$ whereas $\Omega_{\max}$ is set to 0.01, 0.1, 10, and 0.1 for four datasets after grid-searching over

$$[10^{-4}, 10^{-3}, \cdots, 10^2].$$

## B   Related Work

Class-IL strategies, according to what we presented in the previous section, ideally have to tackle the two challenges, TC [16] and CF [1]; these strategies can be divided into four categories: (i) regularization, (ii) bias-correction, (iii) replay, and (iv) generative classifier.

## B.1 Regularization strategy

Regularization is a popular class-IL strategy effective for CF: in order to tackle CF, regularization aims to minimize modifications to parameters lest they cause the model to forget previously learned tasks [27]. The example schemes of this strategy are Elastic Weight Consolidation (EWC) [1], Synaptic Intelligence (SI) [2], and Bayesian inference [28, 29, 1]. Regularization has shown promising results when paired with the rehearsal strategy or with other schemes such as attention [30, 31, 32]. However, for class-IL, they are ineffective alone because they only mitigate CF not TC [33, 34, 35].

## B.2 Bias-correction strategy

Bias-correction strategies are inspired by the observation that when a model is trained in the class-IL setting, it tends to predict only the most recently seen tasks [12, 36]. It is stated that such phenomenon is because the final layer of the neural networks becomes biased towards the latest classes. For that, there have been proposed few class-IL schemes that attempt to remedy this bias by equalizing the values of the final layer's biases of all classes [37]. The exemplar schemes of this strategy are CopyWeights with Re-init (CWR) [20] and its improved version CWR+ [21]. In order to tackle the issues of CWR and CWR+ pertaining to the freezing of the parameters of all hidden layers after the first task and therefore enabling representation learning, the scheme AR1 [11] was proposed, which follows the approach of CWR+. However, AR1 refuses to freeze the hidden layers. Instead, AR1 regularizes the hidden layers following an enhanced version of SI [38]. Bias-correction slightly ameliorates TC; while it does not help with CF.

## B.3 Replay strategy

The replay strategy is based on re-visiting previous data samples. There are two types of replay strategies: (a) coreset/exemplar replay [39, 37, 40] and (b) generative replay [8, 41]. Coreset replay [42, 38] is adopted when it is feasible to store data corresponding to the previous tasks. These stored data samples, i.e., coreset data, are used during training of the new tasks to replay previous tasks, thereby mitigating TC/CF [43, 44]. iCaRL [7] leverages coreset replay: this scheme adopts a neural network for representation learning of the features while does classification via computing the distances of data samples to the centroids pertaining to different classes in the latent space, where the class centroids are computed thanks to the stored data. To prevent the feature extractor network from forgetting the previously learned tasks, iCaRL adopts replay: it replays not only the stored data with the current task but also augments it with a modified form of distillation loss [45, 46].

The second type of the replay strategy, generative replay, is employed if it is not possible to store raw data [47, 6]; alternatively, it replays generated pseudo-data via generative models instead of raw data [18, 5]. This category has been shown to be effective for toy datasets with small input dimensions and unconvoluted patterns; nevertheless, it struggles with problems having more sophisticated input patterns and high-dimensions, which is the case for natural images, unfortunately [18]. Recent generative replay works relying on pre-trained feature extractors which benefit from a long non-incremental initialization training sessions [13, 48] have shown improvements in performance on class-IL scenario with natural images.

## B.4 Generative classifier strategy

We categorize the entire class-IL strategies into two types: discriminative and generative. All the schemes (including generative replay) in the above three categories explained so far count as discriminative classifiers because eventually a discriminator performs the classification—despite using generators for replay. However, the last (the fourth) category of class-IL strategies, generative classifier, performs classification only and directly using generative modeling. In [4], authors proposed a novel rehearsal-free generative classifier strategy for tackling TC/CF (delivering state-of-the-art performance); the scheme does energy-based modeling [36] via generative modeling using Variational AutoEncoders (VAEs) [49] and importance sampling [50]. Generative classifier is a unique treatment of class-IL problem: it entirely prevents CF via model isolation, but its strength is in using generative modeling (for classification), which turns out to be immune to the TC problem unlike discriminative modeling as we will prove.

There are three other advantages for generative classifier [4]: (i) it is capable of operating in the task-free setting [8, 9] where task-ID is unavailable at both training/test time. (ii) It enables single-class-learning: learning in scenarios where only one class is available to be learned [4]. (iii) Another advantage of generative classifier over generative replay [18] is that the former is *half* as simple as the latter: in generative classifier a dataset is used to train generative models *only* and *directly* for classification. By contrast, generative replay uses the data to train an *intermediary* generative model which is used to train a discriminative model in order for classification. Thus, generative classifier has to train once at each session, requiring two times less computation than generative replay. SLDA [10] (popular in data mining [51, 52]) is thought to be another form of generative classifier [4]; however, it prevents representation learning.

