# OpenReview forum: "Task Confusion and Catastrophic Forgetting in Class-Incremental Learning: A Mathematical Framework for Discriminative and Generative Modelings"
_NeurIPS.cc/2024/Conference — NeurIPS 2024 poster_

### Official Review · Reviewer_EnNf · 2024-07-05

**Soundness:** 3
**Presentation:** 2
**Contribution:** 4
**Rating:** 7
**Confidence:** 4

**Summary:**

This paper tries to address and analyze the challenge of task confusion (TC) in class-incremental learning (class-IL). This paper proposes the Infeasibility Theorem that demonstrates that achieving optimal class-IL through discriminative modeling is impossible due to TC, even if CF is prevented​. It further proposes the Feasibility Theorem, which shows that optimal class-IL can be achieved with generative modeling, provided CF is prevented. The authors further emprically assess their theorem with traditional class-IL strategies, including regularization, bias-correction, replay, and generative classifier.

**Strengths:**

The writing of paper is clear.
The paper focuses on an interesting view: task confusion and proposes rigorous theorem to analyze it.
The theoretical contribution of this paper is significant. Researchers can use the theorem to guide their method desgin.
The authors further assesses traditional continual learning strategies from the view of task confusion.

**Weaknesses:**

1) I recommend that the authors clarify their theorems, experimental results, and contributions in the introduction section. Additionally, the method comparison should be detailed in the related works section.
2) The authors should give more details about discrimative/generative modeling and related continual learning strategies. Some fresh readers may not understand it.

**Questions:**

See the weaknesses.

**Limitations:**

I do not see any potential negative societal impact of their work.

---

> ### Author Rebuttal · Authors · 2024-08-04
>
> We appreciate the thoughtful review and valuable feedback you provided on our submission. Your constructive insights are greatly appreciated, and we are committed to addressing the points you've raised.
>
> In the revised paper, the following paragraph will be included in the introduction that clarifies the progression of our theoretical results (our contributions):
>
> "For discriminative modeling, Lemmas 1 and 2 form the groundwork that leads to Theorem 1 and its subsequent Corollaries 1 through 5. For generative modeling, Lemma 3 underpins Theorem 2 and Corollary 6. Furthermore, Hypothesis 1 is derived from the principles outlined in Lemma 1."
>
> In the related work section of our revised paper, we will include a table that compares various methods based on their ability to mitigate task confusion and catastrophic forgetting. This table will detail the qualitative attributes of each method and identify whether they utilize discriminative or generative modeling approaches, based on their documentation in the literature.
>
> The following paragraph will be incorporated in the revised paper to give more details about discriminative/generative modeling:
>
> "Since the advent of AlexNet, the neuroscience community has been skeptical of the deep learning community's discriminative modeling approach for classification [1]. They argue that humans do not learn p(y|x) for classification (discriminative modeling); instead, humans learn p(x) which is generative modeling. The primary issue with discriminative modeling is shortcut learning [1], a concern that has recently gained more attention within the deep learning community [2]. In this work, we investigate how shortcut learning can particularly hinder class-incremental learning. We discuss how shortcut learning in discriminative modeling leads to task confusion and argue that generative modeling, in principle, addresses this issue."
>
> To give more detail about related continual learning strategies, we will provide a taxonomy table that exhaustively outlines the related work in continual learning. Also, we will include formal proofs for all of the theoretical contributions in the revised paper. For Lemmas 1 and 2, we will include the proofs submitted as a global rebuttal. The rest of the formal proofs are also available but not submitted due to the lack of space.
>
> We would be glad to answer any further questions the reviewer may have on this matter.
>
> Thank you for reading our response.
>
> [1] Geirhos, Robert, et al. "Shortcut learning in deep neural networks." Nature Machine Intelligence 2.11 (2020): 665-673.
>
> [2] Yang, Wanqian, et al. "Chroma-VAE: Mitigating shortcut learning with generative classifiers." Advances in Neural Information Processing Systems 35 (2022): 20351-20365.

---

### Official Review · Reviewer_fG2K · 2024-07-06

**Soundness:** 3
**Presentation:** 2
**Contribution:** 3
**Rating:** 6
**Confidence:** 2

**Summary:**

This paper presents a novel mathematical framework for class-incremental learning and prove the Infeasibility Theorem, showing optimal class-incremental learning is impossible with discriminative modeling. While generative modeling can achieve optimal class-incremental learning with the Feasibility Theorem. The analysis suggests that adopting generative modeling is essential for optimal class-incremental learning.

**Strengths:**

1. The motivation is strong and clear, and the importance of such theoretical framework is significant.
2. The proposed framework is insightful and well-structured.

**Weaknesses:**

1. The proofs in appendices are informal with few equations.
2. In the appendix, Figure F.1 is missing, only the text *SIC.pdf* is presented.

**Questions:**

1. Can you elaborate more on how the generative classifier promises optimal class-incremental learning?

**Limitations:**

The proofs should be more formal to make the theoretical framework complete.

---

> ### Author Rebuttal · Authors · 2024-08-04
>
> We're grateful for your thoughtful review and the insightful feedback on our submission. Your constructive comments are highly valued, and we aim to respond to the issues highlighted.
>
> In the revised manuscript, we will include the SIC.pdf file (also submitted as a global rebuttal). The formal proofs of Lemma 1 and Lemma 2 are submitted as a global rebuttal. The rest of the formal proofs (for Corollary 1, Corollary 2, Theorem 1, Lemma 3, and Theorem 2 ) could not be submitted due to lack of space but are available and we are happy to provide the formal proofs upon the reviewer's request during the discussion period.
>
>
> To answer the question of how the generative classifier promises optimal class-incremental learning we would like to invite the reviewer to read the following paragraph that encompasses the big picture of our work.
>
> Big Picture. Since the advent of AlexNet, the neuroscience community has been skeptical of the deep learning community's discriminative modeling approach for classification [1]. They argue that humans do not learn p(y|x) for classification (discriminative modeling); instead, humans learn p(x) which is generative modeling. The primary issue with discriminative modeling is shortcut learning [1], a concern that has recently gained more attention within the deep learning community [2]. In this work, we investigate how shortcut learning can particularly hinder class-incremental learning. We discuss how shortcut learning in discriminative modeling leads to task confusion and argue that generative modeling, in principle, addresses this issue. We would be glad to answer any further questions the reviewer may have on this matter.
>
> Thank you for reading our response.
>
> [1] Geirhos, Robert, et al. "Shortcut learning in deep neural networks." Nature Machine Intelligence 2.11 (2020): 665-673.
>
> [2] Yang, Wanqian, et al. "Chroma-VAE: Mitigating shortcut learning with generative classifiers." Advances in Neural Information Processing Systems 35 (2022): 20351-20365.

---

> > ### Comment · Reviewer_fG2K · 2024-08-12
> > **Thanks for the Response**
> >
> > The authors' response resolves my concerns, I decide to keep my rating.

---

### Official Review · Reviewer_PJ6U · 2024-07-09

**Soundness:** 2
**Presentation:** 3
**Contribution:** 2
**Rating:** 5
**Confidence:** 2

**Summary:**

The paper proposes a Mathematical Framework for class-incremental learning in discriminative and generative modelings, presenting a Infeasibility Theorem for discriminative models and Feasibility Theorem for generative modelings.

**Strengths:**

The paper is easy to understand. It offers a Mathematical Framework for modeling class-IL problem.

**Weaknesses:**

1. spelling error: Bias-Correction Impotence Corollary, "impotence" means "importance"? and in Corollary 1 (Catastrophic Forgetting) miss a inequality sign, and the same problem in corollary 2 ?
2. The Infeasibility theorem is hard to understand, the grammar seems wrong? "The CF-optimal class-IL model in Definition 3 is not be optimal loss are incompatible."
3. proof in Appendix E is unreadable, eg. “SIC.pdf”?

The problem modeling of class IL is good, however, the conclusion and corresponding prove needs more explanation, which is hard to follow, especially the prove in Appendix E. The most important prove part of this paper is put into the Appendix E, while Appendix E is too simple with only oral expression.

**Questions:**

No more questions.

**Limitations:**

The limitations are proposed in the paper.

---

> ### Author Rebuttal · Authors · 2024-08-04
>
> Thank you for taking the time to review our submission and for providing valuable feedback. We appreciate the constructive comments and would like to address the points raised.
>
> This response is organized as follows: first, we specifically address the reviewer's comments (Responses). Second, we provide a paragraph to justify the existence and significance of this work by discussing the bigger picture and how it benefits the class-incremental learning community (Big Picture). We believe the reviewer will find these explanations useful and interesting, and we hope this will persuade the reviewer to reconsider their score.
>
> Responses. We would like to clarify that the word “impotence” in “Bias-Correction Impotence Corollary” is not a spelling error. The term “impotence” meaning ineffectiveness is intentionally used to convey the ineffectiveness of Bias-Correction strategies to overcome the problem of Task Confusion (more detail on this in the final paragraph in Big Picture).
>
> We are grateful to the reviewer for noting the missing inequality signs in Corollaries 1 and 2. We will correct these typos in the revised manuscript. Additionally, Theorem 1 will become as follows:
>
> The CF-optimal class-IL model in Definition 3 is not optimal if the entire loss and the diagonal loss are incompatible:
>
> $$
> \sum_{i=1}^{T} \sum_{j=1}^{T} \left| \boldsymbol{P}_{ij} (\boldsymbol{\theta}) \right| \nparallel \sum_{i=1}^{T} \left| \boldsymbol{P}_{ii} (\boldsymbol{\theta}) \right|.
> $$
>
> In the revised manuscript, we will include the SIC.pdf file (also submitted as a global rebuttal). The formal proofs of Lemma 1 and Lemma 2 are submitted as a global rebuttal. The rest of the formal proofs (for Corollary 1, Corollary 2, Theorem 1, Lemma 3, and Theorem 2 ) could not be submitted due to lack of space but are available and we are happy to provide the formal proofs upon the reviewer's request during the discussion period.
>
> Big Picture. Since the advent of AlexNet, the neuroscience community has been skeptical of the deep learning community's discriminative modeling approach for classification [1]. They argue that humans do not learn p(y|x) for classification (discriminative modeling); instead, humans learn p(x) which is generative modeling. The primary issue with discriminative modeling is shortcut learning [1], a concern that has recently gained more attention within the deep learning community [2]. In this work, we investigate how shortcut learning can particularly hinder class-incremental learning. We discuss how shortcut learning in discriminative modeling leads to task confusion and argue that generative modeling, in principle, addresses this issue. We would be glad to answer any further questions the reviewer may have on this matter.
>
> Thank you for reading our response.
>
> [1] Geirhos, Robert, et al. "Shortcut learning in deep neural networks." Nature Machine Intelligence 2.11 (2020): 665-673.
>
> [2] Yang, Wanqian, et al. "Chroma-VAE: Mitigating shortcut learning with generative classifiers." Advances in Neural Information Processing Systems 35 (2022): 20351-20365.

---

> > ### Comment · Reviewer_PJ6U · 2024-08-13
> >
> > Thanks for your response. As a out-of-domain reviewer, I will raise my score to 5.

---

> > > ### Author Response · Authors · 2024-08-13
> > > **Thanks for your feedback**
> > >
> > > We appreciate your engagement. It seems like the score is still 4. Thank you very much.

---

### Author Rebuttal · Authors · 2024-08-04

Formal proofs for Lemma 1 and Lemma 2:

(Formal proofs of Corollary 1, Corollary 2, Theorem 1, Lemma 3, and Theorem 2 are available and will be included in the revised version but not here due to lack of space.)

We'd like to provide a more formal and rigorous proof for Lemma 1, for how the loss function of an $N$-way classifier is mathematically equivalent to the combined loss functions of $\binom{N}{2}$ binary classifiers. This approach ensures that we account for each pair of classes only once, thereby avoiding any overlap.
Starting with a simple scenario where $N=2$, the loss function is defined as:
$$
I_{\boldsymbol{\theta}} = \int_{\mathcal{X} \times \{1, 2\}} v(f_{\boldsymbol{\theta}}(x), y) p(x, y) \, dx \, dy
$$
This setup is naturally a binary classifier, focusing solely on one class pair.
When considering $N=k$, with $k \geq 2$, the loss function can be expanded to:
$$
I_{\boldsymbol{\theta}} = \frac{1}{k-1} \sum_{i=1}^{k} \sum_{j=i+1}^{k} \int_{\mathcal{X} \times \{i, j\}} v(f_{\boldsymbol{\theta}}(x), y) p(x, y) \, dx \, dy
$$
Here, each class combination $(i, j)$, where $i < j$, is included once, optimizing the calculation and removing redundancy.
For example, when $N=3$, we break it down as follows:
$$
I_{\boldsymbol{\theta}} = \frac{1}{2} \left(
\int_{\mathcal{X} \times \{1, 2\}} v(f_{\boldsymbol{\theta}}(x), y) p(x, y) \, dx \, dy +
\int_{\mathcal{X} \times \{1, 3\}} v(f_{\boldsymbol{\theta}}(x), y) p(x, y) \, dx \, dy +
\int_{\mathcal{X} \times \{2, 3\}} v(f_{\boldsymbol{\theta}}(x), y) p(x, y) \, dx \, dy
\right)
$$
Introducing an additional class, $k+1$, results in more binary comparisons:
$$
I_{\boldsymbol{\theta}} = \frac{1}{k} \left(
\sum_{i=1}^{k} \sum_{j=i+1}^{k} \int_{\mathcal{X} \times \{i, j\}} v(f_{\boldsymbol{\theta}}(x), y) p(x, y) \, dx \, dy +
\sum_{i=1}^{k} \int_{\mathcal{X} \times \{i, k+1\}} v(f_{\boldsymbol{\theta}}(x), y) p(x, y) \, dx \, dy
\right)
$$
Conclusively, for any $N$:
$$
I_{\boldsymbol{\theta}} = \frac{1}{N-1} \sum_{i=1}^{N} \sum_{j=i+1}^{N} \int_{\mathcal{X} \times \{i, j\}} v(f_{\boldsymbol{\theta}}(x), y) p(x, y) \, dx \, dy
$$
This equation confirms that the loss function for an $N$-way classifier replicates that of several binary classifiers, with each class pair uniquely represented.


**************************************

For Incompatibility Lemma (Lemma 2), which addresses the relationship between the minimizers of two incompatible functions $f(x)$ and $g(x)$, denoted as $f(x) \nparallel g(x)$. This lemma posits that the minimizer of the sum $f(x) + g(x)$ is distinct from the minimizers of $f(x)$ and $g(x)$ individually, indicated by:
$$
x^* \neq x^f, x^g, \quad x^* = \arg\min_{x} f(x) + g(x), \quad x^f = \arg\min_{x} f(x), \quad x^g = \arg\min_{x} g(x).
$$

To begin, we define two functions as incompatible if the minimization of one does not necessarily imply the minimization of the other. In other words, their minimizers do not coincide. Suppose, for contradiction, that the minimizer $x^f$ of $f(x)$, is also the minimizer of the combined function $f(x) + g(x)$. This would imply:
$$
x^f = \arg\min_{x} f(x) = \arg\min_{x} \left(f(x) + g(x)\right).
$$

Given that at $x^f$, the derivative of $f(x)$ with respect to $x$ must equal zero:
$$
\frac{df(x)}{dx}\Bigg|_{x = x^f} = 0.
$$

If $x^f$ were also a minimizer of $f(x) + g(x)$, the derivative of the sum at $x^f$ would similarly vanish:
$$
\frac{d}{dx}\left(f(x) + g(x)\right)\Bigg|_{x = x^f} = 0.
$$

This would suggest that the derivative of $g(x)$ at $x^f$ must also equal zero, leading to:
$$
\frac{dg(x)}{dx}\Bigg|_{x = x^f} = 0.
$$

However, given that $f(x)$ and $g(x)$ are incompatible, $\frac{dg(x)}{dx}\Bigg|_{x = x^f} \neq 0$, indicating that there exists a direction opposite to the gradient of $g(x)$ at $x^f$ which can further decrease $f(x) + g(x)$. This contradiction shows that $x^f$ cannot be the minimizer of $f(x) + g(x)$.

Applying a symmetric argument for $x^g$, suppose $x^g$ is both the minimizer of $g(x)$ and $f(x) + g(x)$:
$$
x^g = \arg\min_{x} g(x) = \arg\min_{x} \left(f(x) + g(x)\right).
$$

Following the same logic as before, we derive that the derivative of $f(x)$ at $x^g$ should be zero, leading to:
$$
\frac{df(x)}{dx}\Bigg|_{x = x^g} = 0.
$$

Yet, since the functions are incompatible, $\frac{df(x)}{dx}\Bigg|_{x = x^g} \neq 0$, reinforcing our contradiction and showing that $x^g$ is not the minimizer of the combined function either.

In conclusion, $x^*$, the minimizer of $f(x) + g(x)$, is distinct from both $x^f$ and $x^g$, illustrating that when functions $f(x)$ and $g(x)$ are incompatible, their individual minimizers cannot optimize the combined function.

---

### Decision · Program_Chairs · 2024-09-25

**Decision:**

Accept (poster)

**Comment:**

Overall the contribution provided by this work is interesting and of value to the community. And the reviewers seem to agree on this through the scores and comments. I think the formalism around task confusion and task forgetting is quite a useful tool to understand potential subtleties of continual learning. However I have to emphasize that having a rough proof (sketch?) in the original manuscript and completing this sketch in the rebuttal is not how the system is meant to work. Overall I think there is substantial content that is coming through the rebuttal, that needs to be fully incorporated in the camera-ready paper and I urge the authors to make sure this will be the case. This is somewhat problematic, as the content in the rebuttal has not been reviewed to the same level as that of the original manuscript (e.g. checking step by step the correctness of the proofs). However given the nature of the lemmas, and the original proofs, after inspecting the rebuttal, I do believe the proofs are correct, and the paper incorporating all the content in the rebuttal is a worthy paper and of interest to the community. So I think the paper should be accepted.
But please include all details in the camera-ready and write-up improvement discussed in the conversation with the reviewers.